# Establishment of an LC-MS/MS Method for the Determination of 45 Pesticide Residues in Fruits and Vegetables from Fujian, China

**DOI:** 10.3390/molecules27248674

**Published:** 2022-12-08

**Authors:** Kunming Zheng, Xiaoping Wu, Jiannan Chen, Jinxing Chen, Wenhao Lian, Jianfeng Su, Lihong Shi

**Affiliations:** 1Fujian CCIC-Fairreach Food Safety Testing Co., Ltd., Fuzhou 350015, China; 2Key Laboratory of Molecular Biology of Crop Pathogens and Insects, Zhejiang Institute of Pesticide and Environmental Toxicology, College of Agricultural and Biotechnology, Zhejiang University, Hangzhou 310058, China

**Keywords:** pesticide residue monitoring, fruits and vegetables, QuEChERS, LC-MS/MS

## Abstract

Pesticide residues in food have become an important factor seriously threatening human health. Therefore, this study was conducted to determine the pesticide residues in fruits and vegetables commonly found in Fujian, China, with the aim of constructing a simple and rapid method for pesticide residue monitoring. We collected 5607 samples from local markets and analyzed them for the presence of 45 pesticide residues. A fast, easy, inexpensive, effective, robust, and safe (QuEChERS) multi-residue extraction method followed by liquid chromatography equipped with triple-quadrupole mass spectrometry (LC-MS/MS) was successfully established. This 12-min-long analytical method detects and quantifies pesticide residues with acceptable validation performance parameters in terms of sensitivity, selectivity, linearity, the limit of quantification, accuracy, and precision. The linear range of the calibration curves ranged from 5 to 200 mg/L, the limits of detection for all pesticides ranged from 0.02 to 1.90 μg/kg, and the limits of quantification for the pesticides were 10 μg/kg. The recovery rates for the three levels of fortification ranged from 72.0% to 118.0%, with precision values (expressed as RSD%) less than 20% for all of the investigated analytes. The results showed that 726 (12.95%) samples were contaminated with pesticide residues, 94 (1.68%) samples exceeded the maximum residue limit (MRL) of the national standard (GB 2763-2021, China), 632 (11.23%) samples were contaminated with residues below the MRL, and 4881 (87.05%) samples were pesticide residue-free. In addition, the highest number of multiple pesticide residues was observed in bananas and peppers, which were contaminated with acetamiprid, imidacloprid, pyraclostrobin, and thiacloprid.

## 1. Introduction

Pesticide residues in vegetables and fruits are an important indicator of food safety and are closely related to human health, therefore attracting much attention in recent years. Fruits and vegetables are often over-sprayed with pesticides to prevent pests and increase yields, resulting in serious pesticide overloads [1]. Pesticide residues are inevitable after application, but a residual amount exceeding the national maximum residue limit standards will have adverse effects on human and animals or cause poisoning in organisms in the ecosystem through the food chain [2]. The use of pesticides provides unquestionable benefits in increasing agricultural production in order to grow the quantity and quality of food needed to sustain the human population [3]. The global use of pesticides has been documented to be as high as 3.5 million tons [4]. Thus, agricultural products contaminated with pesticide residues are by far considered the most common way for chemical contaminants to reach humans [5]. Food safety is a top priority for public health protection, and ensuring the safety of fresh food is especially important. This is especially true for fruits and vegetables, which are consumed directly without any processing and in the largest quantities [6]. Despite their many advantages, pesticides can also be dangerous and toxic substances that pollute the environment, and their fate and function remain unknown to a considerable extent [7,8].

Liquid chromatography–tandem mass spectrometry (LC-MS/MS) and gas chromatography–tandem mass spectrometry (GC-MS/MS) using the multireactive ion monitoring (MRM) detection mode have been widely used in the detection of pesticide residues in fruits and vegetables [9,10,11,12]. To improve the precision of experimental results, pretreatment purification methods are required due to the presence of hundreds of chemical substances in fruits and vegetables, which can cause significant interference during the detection process. The QuEChERS (Quick, Easy, Cheap, Effective, Robust, Safe) method has earned its place in food analysis as an alternative to classical extraction techniques. Initially, it was used for the effective isolation of veterinary drugs in animal tissues. After realizing its great potential in the extraction of polar and particularly basic compounds, the original QuEChERS method was adapted in 2003 [13] for pesticide residue analysis in plant material, with great success. Today, it has become the main analytical tool in most pesticide monitoring laboratories because it allows one to obtain high-quality results for a wide range of pesticides at the same time, and it presents all the practical advantages expected by laboratories compared to most traditional analytical methods. Li et al. [14] established a simple and effective method based on QuECHERS coupled with GC-MS/MS for the determination of multiclass pesticides in P. notoginseng by optimizing the extraction and cleanup. Tankiewicz et al. [15] optimized the extraction solvent ratios to establish a multi-residue analysis method for 31 pesticides in fresh fruit and vegetables. Lehotay et al. [16] used gas chromatography and liquid chromatography (GC and LC) coupled with mass spectrometry (MS) to compare different QuEChERS conditions, a method was established for the detection of 32 pesticide residues in fruits and vegetables. Zaidon et al. [17] developed sensitive ex-traction methods using QuEChERS and SPE coupled with UHPLC-MS/MS for multi-residue analysis of 13 pesticides in soil and water. However, most of the research only established detection methods for which the detection matrices are singular or time-consuming. In this study, by optimizing instrument conditions and purifying agents for pre-treatment, the detection efficiency can be improved. In addition, more representative samples are tested, which is conducive to a comprehensive understanding of the real situation of pesticide residues on crops and provides a large amount of basic data for risk assessment and safe use of pesticides.

To protect consumer health from unacceptable levels of pesticide residues in food and feed, maximum residue limits(MRLs) (http://down.foodmate.net/standard/sort/3/97819.html. Accessed on 3 March 2021) for pesticides have been developed in China to reduce environmental and health concerns. In this study, a simple method to detect 45 pesticide residues in fruits and vegetables from Fujian Province, China, was applied to understand the status of pesticide residues in fruits and vegetables sold in Fujian in response to social concerns. In combination with the consumption characteristics of Fujian, the analysis of 45 pesticides was carried out on the fruits and vegetables from 2021 to 2022 according to the requirements of the national food safety risk monitoring plan. The results of this study provide a basis for regulatory authorities to carry out targeted supervision of pesticide residues.

## 2. Results and Discussion

### 2.1. Optimization of MS/MS Condition

Each component to be tested was prepared with acetonitrile in a single standard solution with a concentration of about 0.1 mg/L. The mass spectrometry conditions of 45 compounds were optimized in ESI^+^ and ESI^−^ modes, and the best mode for precursor ion response was selected as the final ion source mode. The response of Fipronil and its three metabolites was better under ESI^−^, so the ESI^−^ mode was selected, and the ESI^+^ mode was selected for the other compounds. In the selected ionization mode, the fragment ions were optimized, and the two pairs of ions with the best response intensity were selected as the monitoring ions. The ion with the least interference and the highest response was used as the quantitative ion, and the remaining ions were used as qualitative ions. At the same time, the collision energy of the compound was optimized. The optimal MRM detection parameters for each pesticide were listed in Table 1.

### 2.2. Optimization of the Sample Preparation Method

Considering the ingredients of fruits and vegetables, this study selected anhydrous magnesium sulfate (MgSO_4_**)**, primary secondary amine (PSA) and graphitized carbon black (GCB) as purifiers and optimized their dosage. The experimental results were measured by the number of pesticides whose spiked recoveries of 45 pesticides (the average value of the experiment was repeated three times) were between 70% and 110%. According to previous reports in the literature [13], the influence of the dosage of PSA when the dosage of anhydrous magnesium sulfate was set to150 mg on the purification effect was investigated. Samples of 5, 10, 15, 20, 25, and 30 mg of PSA were added to 1 mL of the extract previously mixed with 20 μg/kg of the target compound. The results (Figure 1) indicated that the recovery rate of each pesticide had little difference with the increase in PSA dosage; when the PSA dosage was greater than 15 mg, the color of the extract gradually became lighter, but there was no obvious difference after the dosage exceeded 25 mg. Therefore, the dosage of PSA was determined to be 25 mg. Under the condition that the dosage of PSA was 25 mg and the dosage of anhydrous magnesium sulfate was 150 mg, the effect of the dosage of GCB on the purification effect was investigated. Samples of 1, 2, 5, 10, and 20 mg of GCB were added to 1 mL of the extract solution in which 20 μg/kg of the target compound was previously added. The results (Figure 1) indicated that the color of the extract became lighter with the increase in the amount of GCB. When the amount of GCB was 5 mg, it was basically colorless and transparent. The recoveries of pesticides with a planar structure similar to GCB, such as emamectin benzoate, acetamiprid, and carbofuran, began to decline. Therefore, the dosage of GCB was determined to be 5 mg. After optimizing the type and content of salt in the salt bag, 25 mg PSA was finally determined, and 5 mg GCB and 150 mg anhydrous MgSO_4_ can guarantee that the recovery of 45 pesticides greater than 70% can be reached.

### 2.3. Method Validation

The quick, sensitive, and robust QuEChERS method was used to extract multiresidue pesticides from the fruit and vegetable samples. According to the EU SANTE/12682/2019 guideline (EU, 2019) [18], the representative matrix was selected as our validation study for the high-water-content commodity group. The results showed that the recoveries of the three fortification levels were between 72.0% and 118.0%, and all the investigated analytes met the standards for quantitative methods of pesticide residues in food (the precision values were less than 20%) (Table 2).

The Linearity was evaluated using calibration curves in different ranges for different pesticide residues (Table 3). The linear range of the calibration curves ranged from 5.0 to 200.0 mg/L. All the pesticide LODs ranged from 0.02 to 1.90 μg/kg, and the pesticides’ LOQs were 10 μg/kg. The determination coefficient varied between 0.99185 and 0.99988, indicating the suitability of the method for pesticide quantification. The instrument responses for the reagent blank and blank control samples were less than 30% of the LOQ [19]. The linearity, LOD, LOQ, precision (RSD), and accuracy (determined by recovery studies) for the different pesticide residues are shown in Table 2 and Table 3. According to the three spiking levels (i.e., 10.0, 20.0, and 100.0 μg/kg), the recovery of the analyzed pesticides ranged from 72.0% for cyromazine to 118.0% for aldicarb. Moreover, the recoveries were all within the appropriate range of the SANTE/12682/2019 guidelines (European Commission, 2019). The matrix-matched calibration method was proposed to minimize the matrix effect. The repeatability of the method was evaluated by calculating the Relative Standard Deviation (RSD), and the results showed that the RSD was 3.3–9.8% at 10.0 μg/kg, 1.9–6.1% at 20.0 μg/kg, and 1.3–5.0% at 100.0 μg/kg.

### 2.4. The Actual Sample Application

The concentrations of the pesticide residues detected in 5607 samples of fruits and vegetables from Fujian Province indicated that 726 samples (12.95%) were found with pesticide residues, of which 94 samples (1.68%) exceeded the maximum residue limit (MRL) of the national standard (GB2763-2021), 632 samples (11.23%) were below the MRL, and 4881 samples (87.05%) were free of pesticide residues (Table 4). Apples, bananas, peppers, grapes, plums, and peaches had higher positive sample rates, with percentages of 28.77%, 26.57%, 23.27%, 22.92%, 18.95%, and 18.05%, respectively as shown in Table 4. The highest percentages of non-compliance with the national food safety standard’s maximum residue limits for pesticides in food (GB2763-2021) were 7.69%, 3.80%, 2.82%, 1.08%, 0.83%, 0.75%, 0.28%, and 0.16%, respectively.

The frequency and ranges of the detectable pesticide residues in the tested commodities were listed (Table 5). The most frequently detected pesticides were clothianidin in pepper (38.40%), acetamiprid in cabbage (44.59%), clothianidin in aubergine (21.21%), clothianidin in cucumber (65.52%), imidacloprid in banana (35.53%), dimethomorph in grape (32.73%), dimethomorph in strawberry (36.36%), carbofuran in cowpeas (36.36%), clothianidin in lettuce (83.33%), carbendazim in peach (66.67%), carbendazim in kiwifruit (100%), carbendazim in leek (100%), carbendazim in plum (77.78%), dimethomorph in tomato (45.45%), and acetamiprid in apple (66.67%). In addition, Acetamiprid, clothianidin, imidacloprid, pyraclostrobin, clothianidin, and carbendazim were found most often in the tested samples (Figure 2). Multiple pesticide residues were most frequently observed in pepper, banana, cowpea, leek, grape, lettuce, and apple (Table 4).

## 3. Materials and Methods

### 3.1. Chemicals and Materials

The pesticide standards (purities in the range 95–99.9%) were purchased from Dr. Ehrenstorfer GmbH (Augsburg, Germany). Acetonitrile (LC-MS/MS grade) was purchased from Merck (Darmstadt, Germany). The syringe filters (nylon, 0.22 µm) and acetic acid, sodium chloride (NaCl), sodium citrate (C_6_H_5_Na_3_O_7_), citric acid (C_6_H_8_O_7_), and anhydrous magnesium sulfate (MgSO_4_) analytical-grade reagents were purchased from Sinopharm Chemical Reagent (Beijing, China). Distilled water was obtained from Watsons Co., Ltd. (Dongguan, China). Primary secondary amine (PSA, 40–60 µm) and graphitized carbon black (GCB, 40–60 µm) were purchased from ANPU Experimental Science and Technology Co., Ltd. (Shanghai, China).

### 3.2. Sample Preparation

In this study, 15 kinds of fresh fruits and vegetables (pepper, cabbage, eggplant, cucumber, banana, grape, strawberry, cowpea, lettuce, peach, kiwifruit, leek, plum, and tomato) were selected as research objects. These fruits and vegetables were collected from Fujian, China, in February 2021 and June 2022. The edible parts of the fruits and vegetables were shrunk and cut up and then fully mixed and ground with a crusher to obtain samples to be tested. Samples were stored at −20 °C.

### 3.3. Preparation of Standard Solutions

The stock standard mixture was obtained by diluting a mixture solution (an appropriate amount taken from all primary solutions which were made in acetone) with acetone to the level of 10 μg/mL and applied for the preparation of working standard solutions. All solutions made as above were stored at −18 °C when not in use.

Matrix-matched standard solutions were prepared as follows: blank samples were treated by the developed preparation method to obtain the extracts, which were dried through nitrogen evaporation, and then 1 mL of the working standard solutions were added with different concentrations separately, shaken, and finally filtered through a 0.22 μm organic membrane to obtain matrix-matched standard solutions of the corresponding concentrations.

### 3.4. UPLC-MS/MS Analysis

The UPLC-MS/MS system comprised an Agilent Series 1290 ultra-performance liquid chromatography system and a 6470A triple quadrupole mass spectrometer. The ZORBAX Eclipse Plus C18 chromatographic column (2.1 mm × 50 mm, 1.8 μm, Agilent) was used to separate the compound. The column temperature was maintained at 40 °C, and the injection volume was 2 μL. The separation of compounds was conducted by a binary solvent (Phase A: 0.1% formic acid–water and Phase B: acetonitrile) in UPLC at a flow rate of 0.3 mL/min. The solvent gradient of 40 pesticides is as follows: 0–2 min 35% B, 2–4 min 35–55% B, 4–7 min 55–98% B, 7–9 min 98% B, 9–10 min 35% B, and 10–12 min 35% B. The electrospray ionization (ESI) of Agilent Jet Steam Technology is used to obtain the mass spectra of compounds. The temperature of drying gas and sheath gas (N_2_, purity > 99.98%) were 320 °C and 350 °C, with the flow of 10 L/min and 11 L/min, respectively. The pressure of the nebulizer was 45 psi. The fragmenter and collision energy were optimized for each standard in the mass spectrometer in both positive and negative multiple reaction monitoring (MRM) modes. The retention time and the MRM parameters of each analyte are listed in Table 1. The total ion chromatograms (TICs) of 45 pesticides are shown in Figure 3.

### 3.5. Sample Pretreatment

First, 10.0 g (±0.1 g) of the homogenized sample was weighed in a 50 mL centrifuge tube and added 10 mL acetonitrile, which was then shaken vigorously for 10 min. Subsequently, 4 g anhydrous MgSO_4_, 1 g NaCl, 0.5 g C_6_H_5_Na_3_O_7_, and 1 g C_6_H_8_O_7_ needed to be added and mixed in a vortex mixer immediately for 1 min, then centrifuged at 4000 r/min for 2 min. A 2 mL aliquot of acetonitrile supernatant was transferred to a new clean 10 mL centrifuge tube, containing 25 mg PSA, 5mg GCB, and 150 mg anhydrous MgSO_4_ as sorbents, then vortexed for 30 s, immediately centrifuged at 15,000 r/min for 2 min, directly filtrated through a 0.22 μm organic membrane, and finally analyzed by LC-MS/MS.

### 3.6. Method Validation Parameters

The performance of the analytical method was evaluated by linearity, limit of detection (LOD), limit of quantitation (LOQ), accuracy, and precision. Linearity for all the target pesticides was evaluated by matrix-matched calibration. Calibration curves were drawn by plotting the relative peak area against the concentration of the corresponding calibration standards at calibration levels of 5, 10, 20, 50, 80, 100, and 200 ng/mL. The LOD was determined as the concentration producing a signal-to-noise ratio of 3, and the LOQ was viewed as the lowest spiking level of the respective pesticides. The accuracy and precision were estimated at 10, 20, and 100 μg/kg for all the analytes in 6 replicates at each level. Mean recovery and relative standard deviation (RSD) were employed to measure the accuracy and precision. Before further extraction, the samples were spiked with the pesticides, and the results from the recovery study were assessed for compliance with the European SANTE/12682/2019 criteria: the average recovery must be in the range of 70–120%, and the relevant RSD must be less than or equal to 20%. All analyses were performed using the same blank.

## 4. Conclusions

In this study, a multiresidue method for the rapid and simultaneous determination of 45 pesticides in fruits and vegetables using the QuEChERS procedure and LC-MS/MS analysis were established. Based on the EU SANTE/12682/2019 guideline (EU, 2019), an internal validation method was developed for the routine analysis of 45 pesticide residues. It is verified that this simple quantitative method for pesticide residue detection has acceptable validation test parameters (linearity, detection limit, quantification limit, accuracy, and precision) and is highly applicable.

Using this method, pesticide residues in fresh fruits and vegetables in Fujian, China, were evaluated. Among the popular fruits and vegetables of Fujian, China, the pesticide residue pollution levels of apples, bananas, peppers, grapes, plums, and peaches are the highest. The most common pesticides residues were detected as follows: clothianidin in pepper (38.40%), acetamiprid in cabbage (44.59%), clothianidin in aubergine (21.21%), clothianidin in cucumber (65.52%), imidacloprid in banana (35.53%), dimethomorph in grape (32.73%), dimethomorph in strawberry (36.36%), carbofuran in cowpeas (36.36%), clothianidin in lettuce (83.33%), carbendazim in peach (66.67%), carbendazim in kiwifruit (100%), carbendazim in leek (100%), carbendazim in plum (77.78%), dimethomorph in tomato (45.45%), and acetamiprid in apple (66.67%).

## Figures and Tables

**Figure 1 molecules-27-08674-f001:**
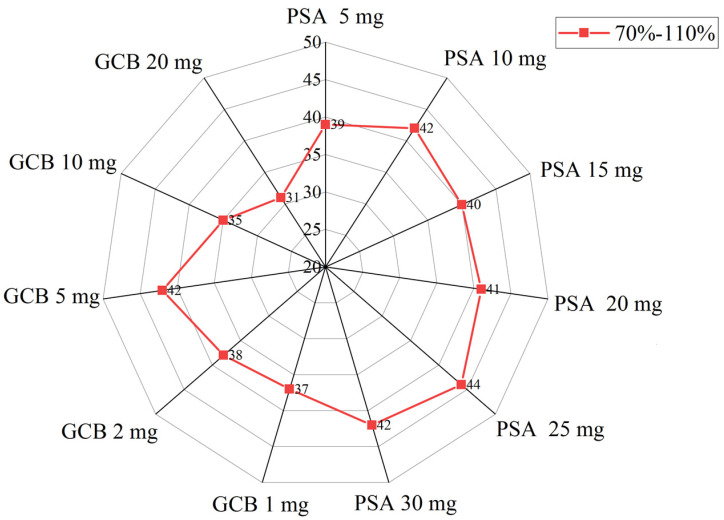
Recovery results for different amounts of sorbent during the purification process (70–110%).

**Figure 2 molecules-27-08674-f002:**
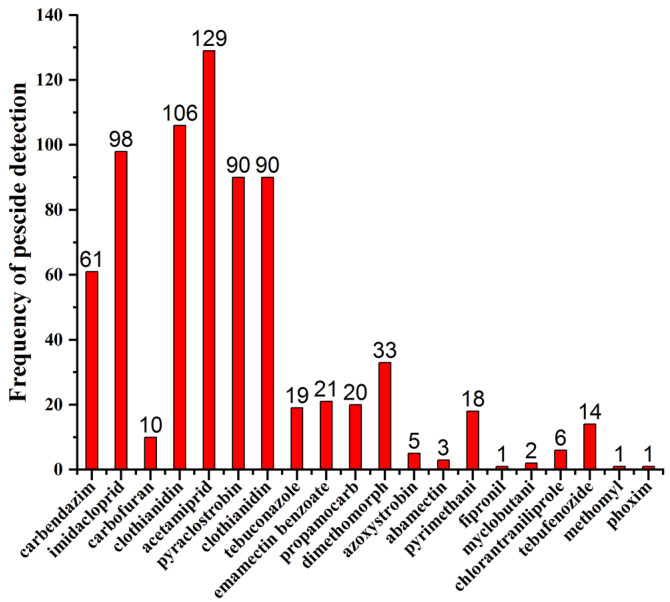
Frequency of the most often detected pesticides in the analyzed samples.

**Figure 3 molecules-27-08674-f003:**
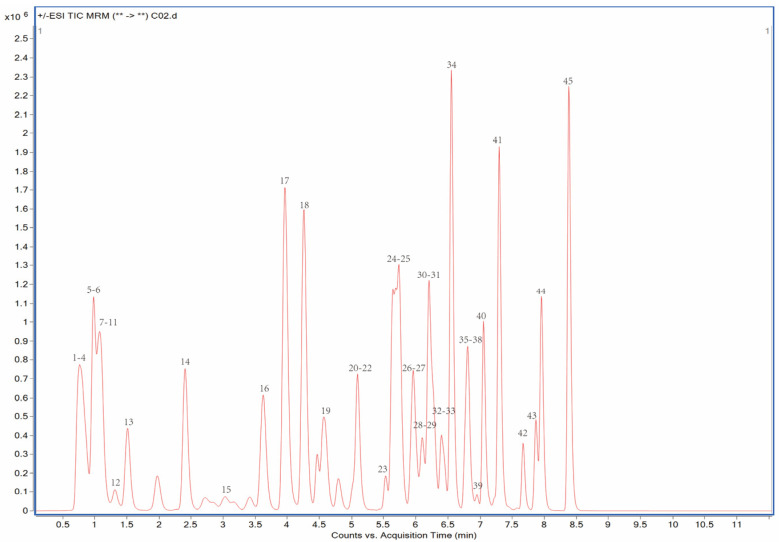
Ultra-performance liquid chromatography multiple reaction mode (UPLC-MRM) chromatogram of 10 μg/L of 45 pesticide standards.

**Table 1 molecules-27-08674-t001:** The MRM acquisition parameters.

NO	Pesticide	Retention Time/min	Quantitative Ion Pair, m/z	Collision Energy/ eV	Qualitative Ion Pair, m/z	Collision Energy/ eV	DP (V)
1	cyromazine	0.737	167.2 > 85.0	25	167.2 > 125.0	20	120
2	propamocarb	0.745	189.0 > 101.9	13	189.0 > 143.9	9	95
3	aldicarb sulfoxide	0.811	207.0 > 131.9	5	207.0 > 88.9	9	60
4	dinotefuran	0.878	203.1 > 129.0	2	203.1 > 157.0	6	85
5	carbendazim	0.991	192.2 > 160.0	15	192.2 > 132.2	32	105
6	chlordimeform	0.998	197.0 > 117.1	38	197.0 > 125.0	30	120
7	aldicarb sulfone	1.004	223.1 > 86.1	9	223.1 > 76.1	9	75
8	thiamethoxam	1.102	292.0 > 211.1	5	292.0 > 181.1	21	82
9	methomyl	1.126	163.0 > 88.0	9	163.0 > 106.0	9	55
10	clothianidin	1.298	250.0 > 169.0	8	250.0 > 131.9	8	110
11	3-hydroxy Carbofuran	1.324	238.0 > 162.8	29	238.0 > 106.9	9	60
12	imidacloprid	1.424	256.0 > 174.7	17	256.0 > 208.6	9	95
13	acetamiprid	1.513	223.0 > 126.0	13	223.0 > 56.0	21	102
14	aldicarb	2.293	213.0 > 89.1	13	213.0 > 116.2	5	65
15	thiophanate-methyl	3.046	343.0 > 151.0	20	343.0 > 311.0	10	120
16	carbofuran	3.644	222.0 > 165.2	9	222.0 > 123.1	21	87
17	carbaryl	3.992	202.0 > 144.8	9	202.0 > 126.9	29	40
18	metalaxyl	4.279	280.0 > 220.0	10	280.0 > 192.0	15	120
19	isoprocarb	4.629	194.1 > 95.0	10	194.1 > 77.0	40	100
20	pyrimethanil	5.041	200.1 > 107.0	25	200.1 > 183.0	25	120
21	chlorantraniliprole	5.094	484.0 > 452.9	17	484.0 > 285.9	50	90
22	dimethomorph	5.188	388.2 > 164.7	25	388.2 > 300.7	9	95
23	myclobutani	5.658	289.1 > 70.1	16	289.1 > 125.1	32	90
24	azoxystrobin	5.712	372.2 > 344.2	18	372.2 > 171.8	42	130
25	fenhexamid	5.836	302.0 > 97.0	25	302.0 > 55.0	30	80
26	tebuconazole	5.946	308.1 > 70.0	40	308.1 > 124.9	47	120
27	flusilazole	5.987	316.1 > 165.0	24	316.1 > 247.1	12	135
28	emamectin benzoate	6.283	886.7 > 158.1	33	886.7 > 126.0	40	160
29	diniconazole	6.277	326.0 > 70.0	30	326.0 > 159.0	25	120
30	propiconazole	6.311	342.0 > 69.0	20	342.0 > 159.0	20	120
31	tebufenozide	6.412	353.3 > 133.1	16	353.3 > 297.2	3	90
32	fipronil	6.462	434.9 > 329.9	13	434.9 > 249.9	25	115
33	isazofos	6.465	314.4 > 120.1	25	314.4 > 162.2	10	100
34	fipronil-desulfinyl	6.588	386.9 > 350.8	13	386.9 > 281.9	33	95
35	fipronil sulfone	6.774	450.9 > 282.0	25	450.9 > 415.0	57	107
36	pyraclostrobin	6.786	388.1 > 193.8	8	388.1 > 163.1	20	140
37	fipronil-sulfide	6.792	418.9 > 382.9	13	418.9 > 261.9	21	100
38	phoxim	6.908	299.0 > 129.0	6	299.0 > 125.1	6	115
39	trifloxystrobin	6.988	409.1 > 186.0	10	409.1 > 206.2	12	140
40	tolfenpyrad	7.073	384.1 > 197.1	25	384.1 > 145.1	30	130
41	epoxiconazole	7.252	330.0 > 121.0	20	330.0 > 141.0	20	120
42	fenpyroximate	7.687	422.2 > 366.1	25	422.2 > 135.1	40	120
43	pyridaben	7.976	365.0 > 147.0	20	365.0 > 309.0	10	80
44	spirodiclofen	7.992	411.1 > 71.2	15	411.1 > 313.0	5	140
45	abamectin	8.496	895.5 > 751.3	45	895.5 > 449.2	50	190

Notes: “D,P” is declustering potential.

**Table 2 molecules-27-08674-t002:** UPLC-MS/MS fortifcation experiments (recovery and repeatability) at 10 μg/kg, 20 μg/kg and 100 μg/kg fortifcation level.

NO	Pesticide	Fortified Level						Intraday Precision /%	Interday Precision /%
		10 μg/kg		20 μg/kg		100 μg/kg			
		Recovery/%	RSD/%	Recovery/%	RSD/%	Recovery/%	RSD/%		
1	cyromazine	79.1	5.0	76.8	1.9	72.0	2.1	2.6	6.3
2	propamocarb	96.3	5.1	94.3	2.1	104.5	2.7	6.1	4.5
3	aldicarb sulfoxide	81.4	4.3	99.8	2.7	98.9	1.6	9.9	9.7
4	dinotefuran	84.0	3.3	99.4	2.6	96.9	1.7	2.6	5.9
5	carbendazim	81.6	3.5	93.7	2.4	94.1	1.7	1.4	3.0
6	chlordimeform	84.2	6.1	95.8	4.3	94.5	2.7	5.5	5.1
7	aldicarb sulfone	85.9	6.5	101.3	3.4	99.2	4.0	1.4	6.7
8	thiamethoxam	83.8	5.2	92.9	2.7	92.5	2.5	1.3	6.3
9	methomyl	89.3	5.9	105.1	3.8	73.9	2.3	4.2	8.9
10	clothianidin	85.1	6.5	101.5	3.5	81.1	2.0	3.5	4.7
11	3-hydroxy Carbofuran	83.9	5.2	105.8	3.7	103.9	2.4	1.6	5.0
12	imidacloprid	84.6	4.8	95.4	3.6	100.2	2.2	5.9	2.7
13	acetamiprid	82.8	6.3	95.6	3.2	106.3	2.2	4.5	6.7
14	aldicarb	80.8	5.4	118.0	3.8	101.6	2.0	1.6	4.7
15	thiophanate-methyl	81.5	5.5	95.7	3.4	101.3	2.5	2.8	8.2
16	carbofuran	97.9	6.6	101.4	3.1	106.1	1.8	1.3	4.5
17	carbaryl	90.6	5.4	90.6	3.3	105.4	1.9	2.0	7.8
18	metalaxyl	98.3	7.0	87.6	3.3	100.1	3.0	3.1	6.5
19	isoprocarb	83.5	5.5	95.9	2.8	102.2	1.7	1.6	4.7
20	pyrimethanil	99.5	5.8	93.1	3.7	102.4	2.0	2.9	4.0
21	chlorantraniliprole	88.5	8.1	95.9	5.1	99.7	1.3	4.4	7.1
22	dimethomorph	84.4	3.6	97.8	3.1	101.5	1.3	2.9	4.0
23	myclobutani	88.0	4.2	105.5	4.0	99.8	2.8	0.8	7.2
24	azoxystrobin	85.6	5.8	83.3	4.0	102.4	2.4	3.9	5.6
25	fenhexamid	83.9	7.9	104.6	4.5	99.0	1.3	2.8	5.9
26	tebuconazole	88.9	5.6	98.4	4.1	99.8	2.4	1.6	5.1
27	flusilazole	93.3	5.8	91.9	2.7	81.7	2.7	1.8	6.7
28	emamectin benzoate	96.6	5.0	98.8	2.4	82.4	3.4	2.1	7.5
29	diniconazole	95.5	4.9	101.6	2.6	82.0	3.4	3.4	3.6
30	propiconazole	98.1	5.8	96.4	2.6	80.3	3.0	3.3	9.2
31	tebufenozide	96.9	5.4	97.0	2.5	82.6	3.1	6.5	4.1
32	fipronil	98.7	5.6	100.7	3.4	84.6	3.3	3.4	3.6
33	isazofos	95.0	5.2	97.8	2.5	80.4	2.9	1.4	5.2
34	fipronil-desulfinyl	95.8	6.7	96.7	4.0	87.9	3.2	5.5	6.2
35	fipronil sulfone	80.8	5.9	103.0	2.9	94.7	2.7	1.4	7.4
36	pyraclostrobin	93.2	8.9	100.3	4.6	96.8	4.8	1.3	6.4
37	fipronil-sulfide	103.5	9.8	96.6	4.1	95.0	5.0	4.2	8.6
38	phoxim	81.0	7.4	98.9	4.3	85.7	3.9	3.5	8.8
39	trifloxystrobin	96.8	5.0	98.5	3.2	74.4	2.5	1.6	4.6
40	tolfenpyrad	98.9	8.9	99.4	4.6	101.4	3.5	5.9	4.2
41	epoxiconazole	99.0	8.0	95.9	4.1	102.2	3.7	4.5	8.3
42	fenpyroximate	98.3	7.9	94.6	3.8	104.6	2.9	1.6	4.6
43	pyridaben	99.3	9.6	96.1	6.1	101.1	4.1	4.1	3.9
44	spirodiclofen	95.1	8.0	96.8	4.8	106.0	3.7	4.4	7.7
45	abamectin	108.4	8.6	95.6	5.5	107.4	3.5	9.8	5.0

Notes: “RSD” is the relative standard deviation.

**Table 3 molecules-27-08674-t003:** Evaluation of the performance of leek sample treatment procedures in terms of coefficient of determination, standard curve, LOD and LOQ.

NO	Pesticide	R^2^	Standard Curve	LOD/(μg/kg)	LOQ/(μg/kg)
1	cyromazine	0.99933	y = 19,681.453054x + 2006.046776	0.13	10.0
2	propamocarb	0.99988	y = 220,790.065387x – 95,578.813453	0.22	10.0
3	aldicarb sulfoxide	0.99967	y = 50,615.675634x − 1884.407035	0.35	10.0
4	dinotefuran	0.99942	y = 49,242.332237x − 4745.757575	0.04	10.0
5	carbendazim	0.99569	y = 554,087.075615x + 549,047.556919	0.02	10.0
6	chlordimeform	0.99954	y = 73,703.996868x + 554.763752	0.03	10.0
7	aldicarb sulfone	0.99569	y = 16,511.982169x + 31,731.162334	0.10	10.0
8	thiamethoxam	0.99874	y = 34,721.499446x + 15,822.961243	0.02	10.0
9	methomyl	0.99941	y = 149,594.578736x + 31,227.223355	0.11	10.0
10	clothianidin	0.99625	y = 7526.907600x + 7103.327426	0.11	10.0
11	3-hydroxy Carbofuran	0.99942	y = 25,246.676150x − 8771.200827	0.09	10.0
12	imidacloprid	0.99983	y = 3497.197561x − 2074.508630	0.17	10.0
13	acetamiprid	0.99973	y = 96,499.483679x − 397.158690	0.02	10.0
14	aldicarb	0.99808	y = 1809.855435x − 895.659126	0.06	10.0
15	thiophanate-methyl	0.99879	y = 84,337.401807x – 122,794.433940	0.39	10.0
16	carbofuran	0.99963	y = 397,830.292305x – 216,360.357125	0.07	10.0
17	carbaryl	0.99914	y = 20,876.837531x + 52.716359	0.79	10.0
18	metalaxyl	0.99944	y = 393,866.314858x + 33,905.052557	0.88	10.0
19	isoprocarb	0.99945	y = 141,887.223197x − 4460.201556	0.02	10.0
20	pyrimethanil	0.99952	y = 72,171.274259x – 16,742.018725	0.12	10.0
21	chlorantraniliprole	0.99831	y = 13,683.711277x + 494.008337	0.18	10.0
22	dimethomorph	0.99977	y = 4748.092851x − 1903.943355	0.02	10.0
23	myclobutani	0.99915	y = 43,932.166296x + 38,545.197963	0.02	10.0
24	azoxystrobin	0.99825	y = 402,097.313091x + 347,915.336374	0.09	10.0
25	fenhexamid	0.99361	y = 5537.204571x + 16,314.322277	0.00	10.0
26	tebuconazole	0.99656	y = 104,452.666634x + 223,024.928605	0.07	10.0
27	flusilazole	0.99839	y = 129,686.641874x + 219,948.779210	0.17	10.0
28	emamectin benzoate	0.99882	y = 45,863.119603x − 2856.709033	0.10	10.0
29	diniconazole	0.99356	y = 70,648.543917x + 277,708.136353	0.05	10.0
30	propiconazole	0.99667	y = 71,671.989052x + 141,207.065166	0.08	10.0
31	tebufenozide	0.99941	y = 132,509.973412x − 759.467957	0.20	10.0
32	fipronil	0.99912	y = 10,218.963617x − 8708.021170	0.01	10.0
33	isazofos	0.99976	y = 49,685.321895x − 9031.684988	0.08	10.0
34	Fipronil-desulfinyl	0.99917	y = 34,340.092591x + 1021.604791	0.11	10.0
35	Fipronil sulfone	0.99982	y = 18,656.031028x − 8501.001907	0.04	10.0
36	pyraclostrobin	0.99916	y = 97,261.255530x + 48361.409204	0.17	10.0
37	Fipronil-sulfide	0.99924	y = 16,252.096423x − 3861.671865	0.01	10.0
38	phoxim	0.99850	y = 5551.637769x + 2639.491110	0.34	10.0
39	trifloxystrobin	0.99896	y = 3193.474716x − 2368.127699	0.07	10.0
40	tolfenpyrad	0.99923	y = 329,537.126816x – 31,214.474639	1.90	10.0
41	epoxiconazole	0.99930	y = 37,413.680279x − 7142.101846	0.18	10.0
42	fenpyroximate	0.99894	y = 181,456.421120x + 160,677.014928	0.41	10.0
43	pyridaben	0.99587	y = 263,408.047994x + 517,359.788618	0.05	10.0
44	spirodiclofen	0.99549	y = 14,281.671476x + 31,928.919622	0.03	10.0
45	abamectin	0.99185	y = 54.098402x + 147.883197	0.13	10.0

Notes: “R^2^” is the coefficient of determination, “LOD” is the limit of detection, “LOQ” is the limit of quantification.

**Table 4 molecules-27-08674-t004:** Monitoring of different pesticide residues in food commodities.

Food Commodity	Number of Samples	Number of Positive Samples	Positive Samples Rate(%)	Pesticides above MRLs Number	Pesticides above MRLs Rate(%)
Pepper	1500	349	23.27	57	3.80
Cabbage	1404	74	5.27	4	0.28
Aubergine	641	33	5.15	1	0.16
Cucumber	419	29	6.92	0	0.00
Banana	286	76	26.57	22	7.69
Grape	240	55	22.92	2	0.83
Strawberry	186	11	5.91	2	1.08
Cowpea	177	17	9.60	5	2.82
Lettuce	164	6	3.66	0	0.00
peach	133	24	18.05	1	0.75
Kiwifruit	107	3	2.80	0	0.00
Leek	102	1	0.98	0	0.00
Plum	95	18	18.95	0	0.00
Tomato	80	9	11.25	0	0.00
Apple	73	21	28.77	0	0.00
Total	5607	726	12.95	94	1.68

**Table 5 molecules-27-08674-t005:** Type of pesticides detected and frequency of detection in tested food commodities.

Food Commodity	Number of Positive Samples	Detected Pesticides	Frequency of Detection (%)	Numberof SAMPLES with Residues > MRL (%)
Pepper	349	carbendazim	13(3.72)	0(0)
		imidacloprid	50(14.33)	0(0)
		carbofuran	3(0.86)	1(0.29)
		acetamiprid	74(3.72)	14(4.01)
		pyraclostrobin	55(15.76)	0(0)
		clothianidin	134(38.40)	39(11.17)
		tebuconazole	6(1.72)	3(0.86)
		emamectin benzoate	6(1.72)	0(0)
		propamocarb	2(0.57)	0(0)
		dimethomorph	2(0.57)	0(0)
		azoxystrobi	2(0.57)	0(0)
		chlorantraniliprole	2(0.57)	0(0)
Cabbage	74	abamectin	1(1.35)	1(1.35)
		emamectin benzoate	11(14.86)	0(0)
		fipronil	1(1.35)	0(0)
		imidacloprid	8(10.81)	1(1.35)
		clothianidin	6(8.11)	0(0)
		tebufenozide	14(18.92)	0(0)
		acetamiprid	33(44.59)	2(2.70)
Aubergine	33	clothianidin	7(21.21)	0(0)
		propamocarb	5(15.15)	1(3.03)
		carbendazim	5(15.15)	0(0)
		imidacloprid	5(15.15)	0(0)
		carbofuran	2(6.06)	0(0)
		emamectin benzoate	2(6.06)	0(0)
		dimethomorph	2(6.06)	0(0)
		phoxim	1(3.03)	0(0)
		chlorantraniliprole	1(3.03)	0(0)
		azoxystrobin	1(3.03)	0(0)
		tebuconazole	1(3.03)	0(0)
		acetamiprid	1(3.03)	0(0)
Cucumber	29	clothianidin	19(65.52)	0(0)
		carbendazim	3(10.34)	0(0)
		propamocarb	2(6.90)	0(0)
		acetamiprid	1(3.45)	0(0)
		imidacloprid	1(3.45)	0(0)
		tebuconazole	1(3.45)	0(0)
		chlorantraniliprole	1(3.45)	0(0)
		dimethomorph	1(3.45)	0(0)
Banana	76	imidacloprid	27(35.53)	15(19.74)
		pyraclostrobin	24(31.58)	0(0)
		clothianidin	19(25.00)	7(9.21)
		carbendazim	6(7.89)	0(0)
Grape	55	dimethomorph	18(32.73)	0(0)
		pyrimethani	17(30.91)	0(0)
		propamocarb	10(18.18)	1(1.82)
		clothianidin	2(3.64)	1(1.82)
		azoxystrobin	2(3.64)	0(0)
		pyraclostrobin	2(3.64)	0(0)
		imidacloprid	2(3.64)	0(0)
		tebuconazole	2(3.64)	0(0)
Strawberry	11	dimethomorph	4(36.36)	1(9.09)
		pyraclostrobin	3(27.27)	0(0)
		carbofuran	1(9.09)	1(9.09)
		clothianidin	1(9.09)	0(0)
		pyrimethani	1(9.09)	0(0)
		acetamiprid	1(9.09)	0(0)
Cowpeas	17	carbofuran	3(17.65)	2(11.76)
		acetamiprid	3(17.65)	0(0)
		emamectin benzoate	2(11.76)	1(5.88)
		abamectin	2(11.76)	1(5.88)
		imidacloprid	2(11.76)	0(0)
		methomyl	1(5.88)	1(5.88)
		myclobutani	1(5.88)	0(0)
		tebuconazole	1(5.88)	0(0)
		clothianidin	1(5.88)	0(0)
		chlorantraniliprole	1(5.88)	0(0)
Lettuce	6	clothianidin	5(83.33)	0(0)
		acetamiprid	1(16.67)	0(0)
Peach	24	carbendazim	16(66.67)	0(0)
		imidacloprid	3(12.50)	0(0)
		pyraclostrobin	2(8.33)	0(0)
		carbofuran	1(4.17)	1(4.17)
		tebuconazole	1(4.17)	0(0)
		chlorantraniliprole	1(4.17)	0(0)
Kiwifruit	3	carbendazim	3(100)	0(0)
Leek	1	carbendazim	1(100)	0(0)
Plum	18	carbendazim	14(77.78)	0(0)
		pyraclostrobin	2(11.11)	0(0)
		tebuconazole	1(5.56)	0(0)
		myclobutani	1(5.56)	0(0)
Tomato	9	dimethomorph	5(45.45)	0(0)
		clothianidin	2(18.18)	0(0)
		pyraclostrobin	2(18.18)	0(0)
		acetamiprid	1(9.09)	0(0)
		propamocarb	1(9.09)	0(0)
Apple	21	tebuconazole	6(28.57)	0(0)
		dimethomorph	1(4.76)	0(0)
		acetamiprid	14(66.67)	0(0)

## Data Availability

Not applicable.

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
