# Peer review of "Establishment of an LC-MS/MS Method for the Determination of 45 Pesticide Residues in Fruits and Vegetables from Fujian, China"

_molecules, 2022, doi:10.3390/molecules27248674_

Round 1

Reviewer 1 Report

The manuscript describe a fast, easy, inexpensive, effective, robust, and safe (QuEChERS) multi-residue extraction method followed by liquid chromatography equipped with triple-quadrupole mass spectrometry(LC-MS/MS) for detecting the pesticide residues in fruits and vegetables commonly found in Fujian, China. However, similar QuEChERS method for detecting the pesticide residues have been widely reported. Therefore, I do not think this work is novel enough for publication in Molecules. Here gives some suggestions for improvement of the manuscript:

1. The title of Investigation and analysis of  pesticide residues in fruits and vegetables from Fujian, China” should be revised.

2. In Abstract, the full name of LODs and “LOQs” should be provided.

3. Introduction section, the previously reported methods for pesticide residues detection should be described. In addition, the innovations of this work should be discussed compared with the previously reported methods.

4. The experimental results of the purification effect of different ratios of GCB, PSA and anhydrous MgSO4 needs to be provided.

5. The experimental results of 2.1. Optimization of GC-MS/MS condition and 2.2. Optimization of the Sample Preparation Method needs to be provided.

6. Line 112, Table 3 should be changed to Table 2. Line 154, Table 3 should be changed to Table 4. Line 212, 10 mg/L?.

7. In Figure 2, 45 pesticides should be marked.

8. Line 117-118, The precision(RSD) and accuracy (determined by recovery studies) for the different pesticide residues is missing in Table 2. Moreover, Intra-assay precision and repeatability for 45 pesticide residues detection should be provided.

9. An extra revision of English is highly desirable.

Author Response

Dear Reviewer:

Thank you so much for your informative and professional comments on our paper. Many thanks for the opportunity to revise the manuscript. We have revised our paper in accordance with your comments.

Point 1: The title of “Investigation and analysis of pesticide residues in fruits and vegetables from Fujian, China” should be revised.

Response 1: Thanks for your suggestion. We have already changed “Investigation and analysis of pesticide residues in fruits and vegetables from Fujian, China” to “Establishment of an LC-MS/MS method for the determination of 45 pesticide residues in fruits and vegetables from Fujian, China.” in the revised paper.

Point 2: In Abstract, the full name of “LODs” and “LOQs” should be provided.

Response 2: Thanks for your suggestion. The full name of “LODs” and “LOQs” was provided in line 19-21.

Point 3: Introduction section, the previously reported methods for pesticide residues detection should be described. In addition, the innovations of this work should be discussed compared with the previously reported methods.

Response 3: Thanks for your suggestion. We have already changed “Nowadays,it has become the main analytical tool in most pesticide monitoring laboratories because it allows to obtain high quality results for a wide range of pesticides at the same time and it presents all the practical advantages expected by laboratories compared to most traditional analytical methods [14-19].” to “Today, it has become the main analytical tool in most pesticide monitoring laboratories because it allows one to obtain high-quality results for a wide range of pesticides at the same time, and it presents all the practical advantages expected by laboratories compared to most traditional analytical methods. Li et al. [14] established a simple and effective method based on QuECHERS coupled with GC-MS/MS for the determination of multiclass pesticides in P. notoginseng by optimizing the extraction and cleanup. Tankiewicz et al. [15] optimized the extraction solvent ratios to establish a multi-residue analysis method for 31 pesticides in fresh fruit and vegetables. Lehotay et al. [16] used gas chromatography and liquid chromatography (GC and LC) coupled with mass spectrometry (MS) to compare different QuEChERS conditions, a method was established for the detection of 32 pesticide residues in fruits and vegetables. Zaidon et al. [17] developed sensitive ex-traction methods using QuEChERS and SPE coupled with UHPLC-MS/MS for multi-residue analysis of 13 pesticides in soil and water. However, most of the research only established detection methods for which the detection matrices are singular or time-consuming. In this study, by optimizing instrument conditions and purifying agents for pre-treatment, the detection efficiency can be improved. In addition, more representative samples are tested, which is conducive to a comprehensive understanding of the real situation of pesticide residues on crops and provides a large amount of basic data for risk assessment and safe use of pesticides.

” in the revised paper.

Point 4: The experimental results of the purification effect of different ratios of GCB, PSA and anhydrous MgSO4 needs to be provided.

Response 4: Thanks for your suggestion. We have already changed “Fruit and vegetable extracts contain a large amount of sugar, pigment, organic acid, and phenolic substances, and the sample matrix is complex. The QuEChERS pretreatment technology is widely used in the analysis of pesticide residues in plant-derived food in recent years, which has the advantages of being fast, simple, and cheap. In this method, PSA is used to adsorb sugars and organic acids, and GCB is used to remove pigment substances. The purification effect of different ratios of GCB, PSA, and anhydrous MgSO4 is investigated by the standard recovery method. The experimental results show that GCB has an obvious adsorption effect on carbendazim and emamectin benzoate with a planar structure. After optimizing the type and content of salt in the salt bag, 25 mg PSA was finally determined, 5 mg GCB and 150 mg anhydrous MgSO4, can guarantee 45 pesticides the recovery can reach more than 70%.” to “Considering the ingredients of fruits and vegetables, this study selected anhydrous magnesium sulfate (MgSO4), primary secondary amine (PSA) and graphitized carbon black (GCB) as purifiers and optimized their dosage. The experimental results were measured by the number of pesticides whose spiked recoveries of 45 pesticides (the average value of the experiment was repeated three times) were between 70% and 110%. According to previous reports in the literature[13], the influence of the dosage of PSA when the dosage of anhydrous magnesium sulfate was set to150 mg on the purification effect was investigated. Samples of 5, 10, 15, 20, 25, and 30 mg of PSA were added to 1 mL of the extract previously mixed with 20 μg/kg of the target compound. The results (Figure 1) indicated that the recovery rate of each pesticide had little difference with the increase in PSA dosage; when the PSA dosage was greater than 15 mg, the color of the extract gradually became lighter, but there was no obvious difference after the dosage exceeded 25 mg. Therefore, the dosage of PSA was determined to be 25 mg. Under the condition that the dosage of PSA was 25 mg and the dosage of anhydrous magnesium sulfate was 150 mg, the effect of the dosage of GCB on the purification effect was investigated. Samples of 1, 2, 5, 10, and 20 mg of GCB were added to 1 mL of the extract solution in which 20 μg/kg of the target compound was previously added. The results (Figure 1) indicated that the color of the extract became lighter with the increase in the amount of GCB. When the amount of GCB was 5 mg, it was basically colorless and transparent. The recoveries of pesticides with a planar structure similar to GCB, such as emamectin benzoate, acetamiprid, and carbofuran, began to decline. Therefore, the dosage of GCB was determined to be 5 mg.” in the revised paper.

Point 5: The experimental results of “2.1. Optimization of GC-MS/MS condition” and “2.2. Optimization of the Sample Preparation Method” needs to be provided.

Response 5: Thanks for your suggestion. The experimental results of “2.1. Optimization of GC-MS/MS condition” was added to Table 1. The experimental results of “2.2. Optimization of the Sample Preparation Method” was provided in the revised paper.

Point 6: Line 112, “Table 3” should be changed to “Table 2”. Line 154, “Table 3” should be changed to “Table 4”. Line 212, 10 mg/L?.

Response 6: Thanks for your suggestion. 1) We have already changed “Table 3” to changed to “Table 3”, and “Table 3” to “Table 4” in the revised paper. 2) We have already changed “10 mg/L” to “10 μg/L” in the revised paper.

Point 7: In Figure 2, 45 pesticides should be marked.

Response 7: Thanks for your suggestion. 45 pesticides were marked in Figure 2 and was shown as follows:

Point 8: Line 117-118, The precision(RSD) and accuracy (determined by recovery studies) for the different pesticide residues is missing in Table 2. Moreover, Intra-assay precision and repeatability for 45 pesticide residues detection should be provided.

Response 8: Thanks for your suggestion. 1) We have already changed “The linearity, LOD, LOQ, precision(RSD) and accuracy (determined by recovery studies) for the different pesticide residues are shown in Table 2.” to “The linearity, LOD, LOQ, precision (RSD), and accuracy (determined by recovery studies) for the different pesticide residues are shown in Table 2 and Table 3.” in the revised paper. 2) Intra-assay precision and repeatability for 45 pesticide residue detection was added in Table 2.

Point 9: An extra revision of English is highly desirable

Response 9: Thanks for your suggestion. We have used the English language editing service provided by MDPI: https://www.mdpi.com/authors/english. The Receipt code was English-edited-55549 and shown as follows:

Reviewer 2 Report

The manuscript presents an investigation and analysis of 45 pesticide residue samples of fruits and vegetables from Fujian, China, with UPLC-MS/MS.

The authors applied a standard for this purpose QuEChERS extraction (optimized for studied samples) and liquid chromatography coupled with MS/MS detection system for an impressive number of 5607 samples. The method of analysis and obtained results are compared with European and China guidelines concerning pesticide residue studies. It is important to control concentration of studied components in available food products due to their negative influence on human health.

The reviewed manuscript is prepared well enough; the methods and the discussion are consequent. However, some shortcomings are noticed after revision article can be recommended for publication in Molecules MDPI Journal.

I would suggest the following revisions to be undertaken:

Abstract. “The linear range of the calibration curves ranged from 5 to 200 mg/L, all the pesticide LODs ranged from 0.02 to 1.90 μg/kg, and the pesticide LOQs ranged were 10 μg/kg.”

10 μg/kg is not a range. Rewrite sentence.

Introduction.

Line 50. Authors use the abbreviation MRM, but full name is not provided.

Line 70. It is not clear what authors understand by “mean vegetables and fruits”

Results and discussion

Lines 89-91. There is a verb missing in the second part of the sentence. Rewrite sentence.

Line 92. Provide the full name of PSA and GCB

Lines 94-96. It is unclear if the results are delivered by authors or by earlier experiments available in the literature. Rewrite sentence.

Lines 96-98. The authors claim to optimize the analytical procedure, but any statistical design of experiment methods is described. It would be more informative if the authors would inform the reader it was one of a time “optimization” or maybe a more advanced procedure was undertaken.

Line 104-107. It is not clear what kind of standards the obtained results met. The SANTE or other? The SANTE guidelines are provided further in the manuscript (lines 236-238), but it is challenging to understand the discussion without such information provided earlier.

Line 114. 10 μg/kg is not a range. Rewrite sentence.

Line 129, Note of Table 2. R2 is not the correlation coefficient. It is coefficient of determination. Change the name or use the proper description, “r” (correlation coefficient is a squared value coefficient of determination).

Line 133. There is a lack of a verb in the first part of the sentence. Rewrite sentence.

Line 138. Misspelling is “peache”; should be “peach”

Line 138. It is not clear what the contamination percentage stands for

The authors use the national standard (GB2736-2021) several times in the manuscript. There is no information on what it stands for. Include information in the most appropriate section of the manuscript.

Materials and methods

Line 177-180. It is unclear what the authors mean by “Shrink and cut edible parts of…” Rewrite sentence.

Table 5. Include information on what “DP” stands for.

Conclusions

Lines 242-243. For 45 pesticide residues?

In the manuscript, editorial errors concerning writing % sign with space between sign and value or Celsius degree sign are present. Correct errors.

Overall revision conclusion is data analysis provided in this manuscript is correct, but providing information on 45 pesticide concentrations in over 5000 food samples calls for more advanced methods offered by chemometrics. Principal component analysis or hierarchical clustering is suitable for this purpose. Including such analysis in the revised manuscript is unnecessary, but in the reviwer’s opinion it would benefit the final version.

Author Response

Dear Reviewer:

Thank you so much for your informative and professional comments on our paper. Many thanks for the opportunity to revise the manuscript. We have revised our paper in accordance with your comments.

Point 1:  Abstract. “The linear range of the calibration curves ranged from 5 to 200 mg/L, all the pesticide LODs ranged from 0.02 to 1.90 μg/kg, and the pesticide LOQs ranged were 10 μg/kg.”

10 μg/kg is not a range. Rewrite sentence.

Response: Thanks for your suggestion. We have already changed “The linear range of the calibration curves ranged from 5 to 200 mg/L, all the pesticide LODs ranged from 0.02 to 1.90 μg/kg, and the pesticide LOQs ranged were 10 μg/kg.” to “ The linear range of the calibration curves ranged from 5 to 200 mg/L, the limits of detection (LODs) for all pesticides ranged from 0.02 to 1.90 μg/kg, and the limits of quantification (LOQs) for the pesticides were 10 μg/kg.” in the revised paper.

Point 2:  Line 50. Authors use the abbreviation MRM, but full name is not provided.

Response: Thanks for your suggestion. We really appreciate the reviewer’s critical and thoughtful comments on MRM, the full name of MRM was provided in line 51-52.

Point 3: Line 70. It is not clear what authors understand by “mean vegetables and fruits”

Response: Thanks for your suggestion. We have already changed “In combination with the consumption characteristics of Fujian, 45 pesticides were carried out on the main vegetables and fruits from 2021 to 2022 according to the requirements of the national food safety risk monitoring plan to provide a basis for the regulatory authorities to carry out targeted supervision.” to “In combination with the consumption characteristics of Fujian, the analysis of 45 pesticides was carried out on the fruits and vegetables from 2021 to 2022 according to the requirements of the national food safety risk monitoring plan. The results of this study provide a basis for regulatory authorities to carry out targeted supervision of pesticide residues.” in the revised paper.

Point 4: Lines 89-91. There is a verb missing in the second part of the sentence. Rewrite sentence.

Response: Thanks for your suggestion. We have already changed “Fruit and vegetable extracts contain a large amount of sugar, pigment, organic acid and phenolic substances, and the sample matrix is complex. ” to “Considering the ingredients of fruits and vegetables, this study selected anhydrous magnesium sulfate, PSA, and GCB as purifiers and optimized their dosage.” in the revised paper.

Point 5: Line 92. Provide the full name of PSA and GCB

Response 5: Thanks for your suggestion. We really appreciate the reviewer’s critical and thoughtful comments on MRM, the full name of PSA and GCB was provided in line 111-112.

Point 6: Lines 94-96. It is unclear if the results are delivered by authors or by earlier experiments available in the literature. Rewrite sentence.

Response 6: Thanks for your suggestion, We have already changed “The experimental results show that GCB has obvious adsorption effect on carbendazim and emamectin benzoate with planar structure.” to “According to previous reports in the literature[13], the influence of the dosage of PSA when the dosage of anhydrous magnesium sulfate was set to150 mg on the purification effect was investigated. Samples of 5, 10, 15, 20, 25, and 30 mg of PSA were added to 1 mL of the extract previously mixed with 20 μg/kg of the target compound. The results (Figure 1) indicated that the recovery rate of each pesticide had little difference with the increase in PSA dosage; when the PSA dosage was greater than 15 mg, the color of the extract gradually became lighter, but there was no obvious difference after the dosage exceeded 25 mg. Therefore, the dosage of PSA was determined to be 25 mg. Under the condition that the dosage of PSA was 25 mg and the dosage of anhydrous magnesium sulfate was 150 mg, the effect of the dosage of GCB on the purification effect was investigated. Samples of 1, 2, 5, 10, and 20 mg of GCB were added to 1 mL of the extract solution in which 20 μg/kg of the target compound was previously added. The results (Figure 1) indicated that the color of the extract became lighter with the increase in the amount of GCB. When the amount of GCB was 5 mg, it was basically colorless and transparent. The recoveries of pesticides with a planar structure similar to GCB, such as emamectin benzoate, acetamiprid, and carbofuran, began to decline. Therefore, the dosage of GCB was determined to be 5 mg.” in the revised paper.

Point 7: Lines 96-98. The authors claim to optimize the analytical procedure, but any statistical design of experiment methods is described. It would be more informative if the authors would inform the reader it was one of a time “optimization” or maybe a more advanced procedure was undertaken.

Response 7: Thanks for your suggestion, We have already changed “The experimental results show that GCB has obvious adsorption effect on carbendazim and emamectin benzoate with planar structure.” to “According to previous reports in the literature[13], the influence of the dosage of PSA when the dosage of anhydrous magnesium sulfate was set to150 mg on the purification effect was investigated. Samples of 5, 10, 15, 20, 25, and 30 mg of PSA were added to 1 mL of the extract previously mixed with 20 μg/kg of the target compound. The results (Figure 1) indicated that the recovery rate of each pesticide had little difference with the increase in PSA dosage; when the PSA dosage was greater than 15 mg, the color of the extract gradually became lighter, but there was no obvious difference after the dosage exceeded 25 mg. Therefore, the dosage of PSA was determined to be 25 mg. Under the condition that the dosage of PSA was 25 mg and the dosage of anhydrous magnesium sulfate was 150 mg, the effect of the dosage of GCB on the purification effect was investigated. Samples of 1, 2, 5, 10, and 20 mg of GCB were added to 1 mL of the extract solution in which 20 μg/kg of the target compound was previously added. The results (Figure 1) indicated that the color of the extract became lighter with the increase in the amount of GCB. When the amount of GCB was 5 mg, it was basically colorless and transparent. The recoveries of pesticides with a planar structure similar to GCB, such as emamectin benzoate, acetamiprid, and carbofuran, began to decline. Therefore, the dosage of GCB was determined to be 5 mg.” in the revised paper. 

Point 8: Line 104-107. It is not clear what kind of standards the obtained results met. The SANTE or other? The SANTE guidelines are provided further in the manuscript (lines 236-238), but it is challenging to understand the discussion without such information provided earlier.

Response 8: Thanks for your suggestion. We have already changed “According to SANTE 2019 guidelines(European Commission, 2019)[20], the representative matrix was selected as our validation study of the high water content commodity group.” to “According to the EU SANTE/12682/2019 guideline (EU, 2019) [18], the representative matrix was selected as our validation study for the high-water-content commodity group.” in the revised paper.

Point 9: Line 114. 10 μg/kg is not a range. Rewrite sentence.

Response 9: Thanks for your suggestion. We have already changed “All the pesticide LODs ranged from 0.02 to 1.90 μg/kg, and the pesticide LOQs ranged were 10 μg/kg.” to “All the pesticide LODs ranged from 0.02 to 1.90 μg/kg, and the pesticides' LOQs were 10 μg/kg.” in line 154-155.

Point 10: Line 129, Note of Table 2. R2 is not the correlation coefficient. It is coefficient of determination. Change the name or use the proper description, “r” (correlation coefficient is a squared value coefficient of determination).

Response 10: Thanks for your suggestion. We have already changed “correlation coefficient” to “coefficient of determination” in line 170 

Point 11: Line 133. There is a lack of a verb in the first part of the sentence. Rewrite sentence.

Response 11: Thanks for your suggestion. We have already changed “Among 5607 fruit and vegetable samples in Fujian Province, 726 samples(12.95%) were contaminated by pesticide residues, 94 samples(1.68%) exceeded the maximum residue limit of the national standard (GB2763-2021), 639 samples(11.40%) were contaminated by pesticide residues below the maximum residue limit, and 4881 samples(87.05%) were free of pesticide residues(Table 3).” to “The concentrations of the pesticide residues detected in 5607 samples of fruits and vegetables from Fujian Province indicated that 726 samples (12.95%) were found with pesticide residues, of which 94 samples(1.68 %) exceeded the maximum residue limit(MRL) of the national standard(GB2763-2021),632 samples(11.23%) were below the MRL, and 4881 samples(87.05%) were free of pesticide residues(Table 4).” in the revised paper.

Point 12: Line 138. Misspelling is “peache”; should be “peach”

Response 12: Thanks for your suggestion. We have already changed “peache” to “peach” in the revised paper.

Point 13: Line 138. It is not clear what the contamination percentage stands for.

Response 13: Thanks for your suggestion. We have already changed “Apple, banana, pepper, grape, plum, and peach had the highest contamination percentages (28.77%, 26.57%, 23.27%, 22.92%, 18.95%, and 18.05%, respectively).” to “Apples, bananas, peppers, grapes, plums, and peaches had higher positive sample rates, with percentages of 28.77%, 26.57%, 23.27%, 22.92%, 18.95%, and 18.05%, respectively as shown in Table 4.” in the revised paper.

Point 14: The authors use the national standard (GB2736-2021) several times in the manuscript. There is no information on what it stands for. Include information in the most appropriate section of the manuscript.

Response 14: Thanks for your suggestion. The information on the national standard (GB2736-2021) was added in lines 82-84.

Point 15: Line 177-180. It is unclear what the authors mean by “Shrink and cut edible parts of…” Rewrite sentence.

Response 15: Thanks for your suggestion. We have already changed “Shrink and cut the edible parts of vegetables and fruits, and then fully mix the vegetables and fruits, and further grind them with a crusher to make samples to be tested, and store them at -20 â„ƒ.” to “The edible parts of the fruits and vegetables were shrunk and cut up and then fully mixed and ground with a crusher to obtain samples to be tested. Samples were stored at -20 â„ƒ.” in the revised paper.

Point 16: Table 5. Include information on what “DP” stands for.

Response 16: Thanks for your suggestion. The note about DP was added in line 107.

Point 17: Lines 242-243. For 45 pesticide residues?

Response 17: Thanks for your suggestion. We have already changed “49 pesticide residues.” to “45 pesticide residues” in line 285.

Point 18: In the manuscript, editorial errors concerning writing % sign with space between sign and value or Celsius degree sign are present. Correct errors. 

Response 18: Thanks for your suggestion. The editorial errors concerning writing % sign with space between sign and value or Celsius degree sign have been corrected in the revised paper.

Point 19: Overall revision conclusion is data analysis provided in this manuscript is correct, but providing information on 45 pesticide concentrations in over 5000 food samples calls for more advanced methods offered by chemometrics. Principal component analysis or hierarchical clustering is suitable for this purpose. Including such analysis in the revised manuscript is unnecessary, but in the reviwer’s opinion it would benefit the final version.

Response 19: Thanks for your suggestion. The current time and conditions are limited, but further research will be done on follow-up topics.

Round 2

Reviewer 1 Report

The present manuscript have gained significant improvement after revision. Here gives some suggestions for further improvement of the manuscript:

1. In Abstract, the “LODs” and “LOQs” should be removed.